# Oxidative Stress in the Brain: Basic Concepts and Treatment Strategies in Stroke

**DOI:** 10.3390/antiox10121886

**Published:** 2021-11-25

**Authors:** Matyas Jelinek, Michal Jurajda, Kamil Duris

**Affiliations:** 1Department of Pathophysiology, Faculty of Medicine, Masaryk University, 62500 Brno, Czech Republic; jelinek.matyas@mail.muni.cz (M.J.); mjuraj@med.muni.cz (M.J.); 2Department of Experimental Biology, Faculty of Science, Masaryk University, 62500 Brno, Czech Republic; 3Department of Neurosurgery, The University Hospital Brno and Faculty of Medicine, Masaryk University, 62500 Brno, Czech Republic

**Keywords:** oxidative stress, free radicals, ROS, RNS, brain, stroke, antioxidants, scavengers

## Abstract

The production of free radicals is inevitably associated with metabolism and other enzymatic processes. Under physiological conditions, however, free radicals are effectively eliminated by numerous antioxidant mechanisms. Oxidative stress occurs due to an imbalance between the production and elimination of free radicals under pathological conditions. Oxidative stress is also associated with ageing. The brain is prone to oxidative damage because of its high metabolic activity and high vulnerability to ischemic damage. Oxidative stress, thus, plays a major role in the pathophysiology of both acute and chronic pathologies in the brain, such as stroke, traumatic brain injury or neurodegenerative diseases. The goal of this article is to summarize the basic concepts of oxidative stress and its significance in brain pathologies, as well as to discuss treatment strategies for dealing with oxidative stress in stroke.

## 1. Introduction to Physiology of Free Radicals

Free radicals are highly reactive molecules because they contain unpaired valence electrons [1]. The most important radicals in biological systems are reactive oxygen species (ROS) and reactive nitrogen species (RNS). The production of free radicals is inevitably associated with metabolism and other enzymatic processes. The major sources of free radicals are potentially mitochondria (intracellular) and inflammation (extracellular). Under physiological conditions these processes are highly regulated. When imbalance with anti-oxidative mechanisms occurs these reactive molecules can cause DNA damage and protein oxidation and lipid peroxidation, leading to cell toxicity and tissue damage [2].

Some of the most relevant types of ROS and RNS are superoxide (O_2_^•−^), hydrogen peroxide (H_2_O_2_), hydroxyl radical (HO^•^), hypochlorous acid (HOCl), nitric oxide (NO) and peroxynitrite (ONOO^-^). Superoxide is the most important ROS in the central nervous system (CNS). Intracellularly, it is generated as a side-product of metabolism, mainly the mitochondrial respiratory chain [3]. Extracellularly, it is generated as a part of the immune response, primarily by NADPH oxidases (NOX), myeloperoxidase (MPO) and xanthine oxidase (XO) [4,5]. Superoxide quickly disintegrates into oxygen and hydrogen peroxide. However, in the presence of redox-active iron, highly reactive hydroxyl radicals are generated from superoxide and hydrogen peroxide in the Fenton and Haber–Weiss reactions.

Nitric oxide (NO) is generated by nitric oxide synthase (NOS). It plays an important role in the immune response, where large amounts of NO are generated by phagocytes [6]. Furthermore, through interaction with soluble guanylyl cyclase, it dilates blood vessels, thus raising blood supply and lowering blood pressure [7]. It is also functions as a retrograde neurotransmitter [8].

There are three isotypes, endothelial NOS (eNOS), inducible NOS (iNOS) and neuronal NOS (nNOS). The two most important isotopes involved in stroke are eNOS and iNOS. In the presence of superoxide highly oxidative peroxynitrite can form.

Although an excessive production of free radicals is considered harmful, regulated production is a normal physiological process that plays a vital role in metabolic processes and in the immune response. Therefore, there must also be some anti-oxidative mechanisms to counter the production of ROS and regulate the oxidative state of the cell (Figure 1).

There are many antioxidant mechanisms in the cell to prevent generation of ROS or to remove them. They can be of internal, or external (nutritional) origin. Some of the most important antioxidants include glutathione (GSH), vitamin C and E, superoxide dismutase (SOD), catalase (CAT), glutathione peroxidase (GSH-Px), heme oxygenase (HO) and glutathione reductase (GR). They contribute to the antioxidant potential of the cell and work as an important cellular defense against ROS and RNS. SOD generates hydrogen peroxide, which is converted by CAT to oxygen and water. GSH-Px uses GSH to reduce hydrogen peroxide to water. HO-1 has antioxidant effects, maintains microcirculation in ischemic injury and has modulatory effects on the cell cycle [9].

## 2. Oxidative Stress

Oxidative stress is described as a state of imbalance between the production of free radicals and their elimination by an organism’s anti-oxidative mechanisms. The base of oxidative phosphorylation (and other catabolic processes) is the transfer of electrons, which is essential for energy release. Electrons travel from one protein complex to the next in the inner mitochondria membrane. Therefore, intermediates in this reaction are, in nature, radicals [10]. However, these intermediates are broken down by subsequent reactions. At the end of the electron transport chain, the final electron acceptor is oxygen, and ultimately water is formed, which is not radical in nature [11]. Thus, it is important these reactions take place from their beginnings to their ends. Mitochondrial oxidative stress arises when these reactions stop halfway in, for example, due to lack of oxygen. In that case, antioxidant mechanisms kick in to scavenge or eliminate free radicals. However, they do not have unlimited capacity and their recycling may be limited by the same factors as mitochondrial dysfunction [12].

The other major physiological source of free radicals is inflammation. Inflammation is part of a complex biological response to harmful stimuli, such as bacteria. Upon contact with pathogens (e.g., bacteria), phagocytes try to entrap and destroy them by complex set of digestive enzymes, ROS and RNS [13]. Oxidative stress plays a dual role in an infection. Free radicals are released from immune cells protect against invading pathogens, but they can also cause tissue damage during inflammation [13]. Superoxide can be produced by neutrophils in phagosomes or in the cell membrane surface through the function of an oxidase enzyme (NADPH oxidase and myeloperoxidase) [14,15]. The excessive production of various ROS and RNS overwhelms endogenous antioxidant defense mechanisms and promotes the development of oxidative state.

Given the immense variety and extent of pro-oxidative and anti-oxidative compounds, attempts to categorize oxidative stress have been made, ranging from physiological oxidative stress to excessive and toxic oxidative overload [16,17].

Oxidative stress plays an important role in a number of pathologies and diseases. One of the basic but often overlooked mechanisms that reinforces oxidative stress is a decrease of antioxidant mechanisms due to a lack of vital nutrients [18,19]. The opposite process would be an increase in the generation of free radicals, which can happen through intracellular mitochondrial dysfunction, or through an extracellular source such as inflammation, as discussed above. The time-course of oxidative stress is initially mitochondria-mediated, for example, due to oxygen and glucose deprivation and following mitochondrial depolarization, and is then xanthine oxidase-mediated, NOS-induced and later NADPH oxidase-mediated [20].

Furthermore, oxidative stress is also involved in aging [21]. According to the oxidative stress theory of aging, age-associated functional losses can be attributed to the accumulation of oxidative damage to DNA, proteins and lipids by free radicals [22].

The consequences of oxidative stress are extensive and affect many cellular processes. Oxidative stress damages all important macromolecules. Lipid peroxidation, protein oxidation and DNA fragmentation can lead to multiple cell signaling effects that can result in the initiation of apoptosis [23]. As already mentioned, the primary site for ROS generation is the mitochondria. They can initiate cell death via cytochrome C release leading to activation of the intrinsic apoptotic pathway [24]. Furthermore, oxidative stress plays an important role in cancer development [25,26].

## 3. Oxidative Stress in the Brain

The brain is especially sensitive to oxidative damage because of its high and specific metabolic activity. High consumption of oxygen, almost exclusive oxidative phosphorylation, no reserves of energy, high concentrations of lipids prone to peroxidation, and high levels of iron, all acting as a pro-oxidant [27]. Neuronal cells are, therefore, highly susceptible to metabolic/ischemic damage and associated oxidative stress.

Lipid peroxidation is the main mechanism of oxidative damage by ROS [23]. Furthermore, initiation of lipid peroxidation develops a positive feedback loop [23]. Reactive species are unstable and react quickly with surrounding molecules. Oxidative stress is, therefore, a very rapid pathology, and it is hard to predict the type of damage.

In stroke, the results of ROS imbalance are considerable and include apoptosis, blood-brain barrier (BBB) disruption, inflammation, edema formation, autophagy and other pathophysiological events [20]. ROS and RNS are generated mainly by microglia and astrocytes [28].

## 4. Antioxidant Treatment Strategies of Oxidative Stress in Stroke

Antioxidant treatment, in general, is not very effective because the consequences of oxidative stress are severe. Associated apoptosis is connected with tissue damage and potentially with an impairment of function, which may be irreversible. The oncological consequences are even more severe.

In physiological conditions in the brain, the continuous production of ROS in various physiological processes is balanced by endogenous antioxidant mechanisms. However, after stroke free radicals’ production is massively increased and overwhelms the natural antioxidant systems. There are three main mechanisms by which antioxidants function (Table 1): (1) inhibition of production of free radicals, (2) scavenging of free radicals and (3) increasing free radicals degradation [29].

### 4.1. Inhibition of ROS-Producing Enzymes

This approach targets the source of ROS by utilizing specific inhibitors of ROS-generating enzymes. The two enzymes that are most often targeted in stroke are NADPH oxidase (NOX) and xanthine oxidase (XO).

NADPH oxidase produces superoxide thus inhibition of NOX has been used as a target for stroke treatment. There are several subtypes of NOX; NOX2 and NOX4 make a major contribution to oxidative stress following stroke [30]. Apocynin has been shown to attenuate brain injury after experimental ischemic stroke and to reduce infarct size and levels of the apoptosis-inducing enzymes Bax and Bcl-2 [31,32]. It seems that apocynin functions as a nonspecific antioxidant, because it also inhibits Rho kinase inhibitor, thus the p47phox subunit cannot migrate to the membrane and the NOX complex cannot assemble [33,34,35]. VAS2870 is a pharmacological inhibitor of NOX and has been shown to decrease infarct volume, oxidative stress, neuronal apoptosis and BBB leakage [36]. Several new small-molecule NOX inhibitors have been recently identified, such as VAS2970, M13, GKT136901, ML090 and MLM171 [37,38,39,40]. They have the potential to reduce ROS production and inhibit brain tissue damage [40]. The NOX inhibitors are non-specific and not isoform selective, although it may not be particularly important in stroke treatment, the development of selective NOX inhibitors could provide information about the precise role of various NOX isoforms.

During ischemia, xanthine dehydrogenase undergoes proteolytical convection to xanthine oxidase (XO) [41]. Ischemia also induces intracellular depletion of ATP, which leads to the accumulation of hypoxanthine and xanthine [41]. They serve as substrates for the ROS-producing XO. Allopurinol and its metabolite oxypurinol have been suggested as scavengers of hydroxyl radicals [42]. Allopurinol was found to reduce the infarct size by approximately 35%, and to have protective effects against neurological impairment and mortality [43,44]. Newly used XO inhibitors with ROS scavenging properties include caffeic acid phenethyl ester (CAPE) [45].

Myeloperoxidase (MPO) is a critical enzyme in oxidative stress and neuroinflammation in pathologies such as stroke. MPO activation catalyzes reaction of chloride and H_2_O_2_ creating hypochlorous acid (HOCl) [46]. Targeting MPO with inhibitors (quercetin, eriodictyol, resveratrol, curcumin, trigonelline, dauricine and many others) reduces brain infatuation and improves neurological outcome [46,47,48,49,50,51,52].

### 4.2. Free Radical Scavengers

Compounds capable of scavenging free radicals have been in development for a long time, however the translation from pre-clinical to clinical trials have, so far, been disappointing.

Tirilazad, an inhibitor of lipid peroxidation, was reported in a pre-clinical study to reduce size of lesions and improve neurological score; nevertheless, the clinicals trials have not been successful [53]. Edaravone, a compound known to scavenge peroxyl, hydroxyl and superoxide radicals, was approved for use in Japan in 2001. Pre-clinical studies showed promising results; however in human clinical trials, the positive results are less clear [54].

### 4.3. Free Radical Degradation

Another important strategy that aims at reducing oxidative stress is to increase levels of the antioxidant superoxide dismutase (SOD). SOD catalyzes the conversion of superoxide to less potent H_2_O_2_ and O_2_. Other enzymes, catalase (CAT) and glutathione peroxidase (GSH-Px), help to eliminate H_2_O_2_, thus improving the effectiveness of SOD.

Of the three SOD isoforms, the SOD1 has been the most studied in experimental stroke treatment. Transgenic mice and rats with SOD1 overexpression shown reduced apoptosis [55,56].

Nitric oxide (NO) is an important vasoactive molecule with broad physiological functions. It is produced by NO synthase (NOS) of which there are three isoforms, inducible NOS (iNOS), endothelial NOS (eNOS) and neuronal NOS (nNOS). NO derived from eNOS can have a neuroprotective effect because it can terminate the chain reactions within lipid peroxidation, while NO derived from iNOS acts as pro-oxidant because it can react with superoxide, forming more potent ONOO^-^ [23,57]. Lubeluzole reduces NO levels and, therefore, also ONOO^-^ levels. Pre-clinical results showed promise; nonetheless, the primary clinical trial was terminated early due to unexplained mortality in a higher-dose group of patients [58,59].

### 4.4. Mitochondrial Targeted Antioxidants

The mitochondrial matrix acts as an important endogenous source of free radicals in ischemic conditions. Oxidative damage in mitochondria can lead to increased ROS production, decreased ATP production and the release of pro-apoptotic signals. The inhibition of mitochondrial respiratory chain complex I has been found to be beneficial in stroke treatment [60].

The use of antioxidants targeting mitochondrial ROS generation is complicated due to the difficulty in achieving the high concentrations of antioxidants necessary in the intracellular space [61]. This problem can be solved by conjugating an antioxidant with a lipophilic cation to promote diffusion and accumulation in the mitochondria. Mitoquinone, a derivative of ubiquinone, is reduced to ubiquinol and has been shown to protect mitochondria from the oxidative damage caused by H_2_O_2_ [62].

### 4.5. Antioxidant Supplementation to Scavenge ROS

Vitamin C (ascorbate, ascorbic acid) is a potent antioxidant. Ascorbate directly scavenges ROS and nitrogen-based radicals. It also upregulates eNOS and downregulates NOX. Its antioxidant properties are intensified in combination with other antioxidants, such as vitamin E. Vitamin C plasma concentrations high enough to be able to effectively scavenge ROS can only be achieved by intravenous supplementation [28]. Higher vitamin C serum concentration was associated with a reduced risk of incidence of ischemic and hemorrhagic stroke [63]. The oral supplementation was shown to slightly reduce lipid peroxidation markers [28].

Vitamin E (α-tocopherol) is also a potent antioxidant with the ability to prevent the propagation of ROS chain reactions, mainly in membrane lipids. Vitamin E (isoforms α-tocopherol and γ-tocopherol) has been shown to reduce infarct volume by 45–55% in transient and permanent models of cerebral ischemia [64,65].

An association between vitamin C and vitamin E plasma concentrations has been shown. Vitamin C has the ability to “recycle” α-tocopherol in lipid bilayers and erythrocytes, thus increasing the vitamin E antioxidant capacity [66]. The brain is a lipid-rich environment, so recycling vitamin E may be an important function of ascorbic acid. When stroke patients were given both vitamin C and vitamin 3 supplements, there was a reduction of lipid peroxidation and exerted anti-inflammatory effects [66].

Resveratrol, a polyphenol that is a major component of red wine, has been found to prevent lipid peroxidation and scavenge ROS [67]. Resveratrol-pretreated mice showed significantly reduced infarct size [68]. Resveratrol-treated mice experienced decreased neuronal death, lower glial activation, lower matrix metalloproteinase 9 (MMP9) and induction to neuroprotective HO-1 [67,69].

N-Acytylcysteine (NAC) has been shown to scavenge ROS and increase the rate of endogenous GSH synthesis [70]. NAC administered prior or after reperfusion was shown to reduce lesion size by 49% and 29% and to have anti-inflammatory effects [71].

Other compounds, such as the vitamin B group, extract from Ginkgo biloba, lazaroids, Edaravone and quercetin, have also been shown to have ROS-scavenging, neuroprotective, anti-inflammatory and antioxidant properties in experimentally induced stroke [28,47,72,73,74].

### 4.6. Antioxidant Treatment of Oxidative Stress in Hemorrhagic Stroke

In the treatment of intracerebral hemorrhage (ICH), several natural compounds have been used (such as pyrroloquinoline quinone, melatonin, sulforaphane and ursolic acid) [75,76,77,78,79]. In ICH animal studies, these compounds exhibited neuroprotective effects by inhibiting oxidative stress, upregulating antioxidant levels and reducing secondary brain injury [75,76,77,78,79]. Furthermore, treatment of ICH using hydrogen and active hydrogen compounds, targeted therapy against oxidative stress signaling pathways, synthetic antioxidant drugs (Edaravone) and targeted gene therapy have also been used [75]. Molecules with antioxidant capacity for alleviating oxidative stress were also used in an experimental treatment of subarachnoid hemorrhage [80,81].

**Table 1 antioxidants-10-01886-t001:** A list of compounds used in antioxidant stroke treatment that were discussed in the article.

Compound	Origin	Stroke Type	Target	Outcome	References
allopurinol + ** ±	synthetic	ischemic stroke	XO	stroke volume and cerebral edema reduction	[44]
ascorbate (vitamin C) ++ * ±	natural	ischemic stroke, hemorrhagic stroke	NOS, NOX, vitamin E, free radical scavenger	reduced risk of stroke; lower levels of peroxidation markers; reduced infarct size	[63]
CAPE + ** ±	synthetic	ischemic stroke	XO	XO inhibition	[45]
curcumin ++ * ±	natural	ischemic stroke	MPO, cytokines	inhibition of NF-κB; reduced inflammation and brain damage	[50]
dauricine + ** ±	natural	ischemic stroke	MPO, cytokines	reduced activity of MPO; reduced inflammation	[51]
Edaravone ++ ** ±	synthetic	ischemic stroke	free radical scavenger	improvement of functional outcome	[54]
GKT136901 + ** ±	synthetic	ischemic stroke	NOX1, NOX2, NOX4, NOX5	NOX inhibition	[37]
lubeluzole ++ * ±±	synthetic	ischemic stroke	NOS	reduced infarct volume	[58,59]
M13 + ** ±	synthetic	ischemic stroke	NOX1, NOX4	NOX inhibition	[39]
melatonin + ** ±	natural	ischemic stroke, hemorrhagic stroke	HO-1	increased HO-1 expression; amelioration of brain edema, BBB impairment, apoptosis and neurological deficits	[78]
mitoquinone + ** ±	synthetic	ischemic stroke	mitochondria	recovery of O_2_consumption and complex I activity	[62]
ML090 + ** ±	synthetic	ischemic stroke	NOX1, NOX4, NOX5	NOX inhibition	[38]
ML171 + ** ±	synthetic	ischemic stroke	NOX1, NOX4, NOX5	NOX inhibition	[38]
N-Acetylcysteine ++ ** ±	natural	ischemic stroke	free radical scavenger, GSH	reduction of infarct volume; reduction of expression of pro-inflammatory cytokines; reduced cell death	[71]
neuroglobin + ** ±	natural	ischemic stroke	mitochondria	improved neurological outcome; reduced hypoxia-induced oxidative stress	[82]
pyrroloquinoline quinone + ** ±	natural	hemorrhagic stroke	NOS, mitochondria	alleviation of hematoma volumes; reduced expansion of brain edema and production of ROS	[77]
quercetin + ** ±	natural	ischemic stroke	MPO, SOD, CAT	reduction of infarct size and MPO levels; increase in SOD and CAT levels	[47]
resveratrol ++ ** ±	natural	ischemic stroke, hemorrhagic stroke	MPO, MMP-9, cytokines	reduction of infarct size, neuronal injury, MPO activity, MMPs; reduction of inflammation	[48,52,67]
sulforaphane + ** ±	natural	ischemic stroke, hemorrhagic stroke	HO-1, NOX, GSH	increase in HO-1; inhibition of NOX	[76]
tirilazad ++ * ±±	synthetic	ischemic stroke	lipid peroxidation	reduced infarct volume; increase death rate	[53]
Trigonelline + ** ±	natural	ischemic stroke	MPO, GSH	reduction of infarction; inhibition of MPO and GSH	[49]
ursolic acid + ** ±	natural	ischemic stroke, hemorrhagic stroke	free radical scavenger	attenuation of cerebral edema, BBB disruption, neuronal cell death and neurological deficit	[79]
VAS2870 + ** ±	synthetic	ischemic stroke	NOX1, NOX2, NOX4, NOX5	NOX inhibition	[36]
α-tocopherol (vitamin E) ++ * ±	natural	ischemic stroke, hemorrhagic stroke	prevent the propagation of ROS chain reaction	reduced risk of ischemic stroke	[65]

+ pre-clinical study; ++ clinical study; * compound not effective in trial; ** compound effective in trial; ± compound under evaluation; ±± compound abandoned based on unfavorable results.

## 5. Combination Therapy in Stroke Treatment

No single antioxidant has been approved for stroke treatment. This is due to many factors, such as a narrow treatment window and dose-limiting toxicity [83]. Multimodal therapy would allow the targeting of multiple pathophysiological mechanisms.

For example, targeting stroke combining stem cell therapy, the inhibition of matrix metalloproteinases and oxidative stress have been found more effective than single or double combinations [84]. Also, combining antioxidant (neuroglobin) with apoptotic inhibitor (c-Jun N-terminal kinase) demonstrated improved outcome compared with a single therapy [82].

In a meta-analysis of 126 stroke treatments on animals, single treatment reduced the infarct size by 20% and the neurological score improved by 12%, whereas a second therapy improved the effectiveness by additional 18% and 25%, respectively [85]. The combination of neuroprotective therapy and antioxidant treatment has significant potential to improve the outcome of stroke patients.

Another possible solution could be the use of nanomaterials, which can deliver drugs, and some even have shown potential in scavenging ROS due to their natures [86]. Generally speaking, nanomaterials are usually 1–500 nm in diameter and can easily be taken up by the cell [87]. Metallic and metal-oxide nanoparticles, carbon-based nanoparticles, liposome nanoparticles and polymeric nanoparticles have all been used for antioxidant therapy in stroke [86]. In ischemic stroke, nanoparticles have been used to scavenge ROS, as carriers to transport free radicals, or as carriers to transport antioxidant enzymes, drugs or genes [86]. Nanoparticle-facilitated delivery of antioxidants was also used in the experimental treatment of aneurysmal subarachnoid hemorrhage in rats [88,89].

Mesenchymal stem cells (MSCs) and secretomes have shown beneficial impact on cerebral ischemic injury. MSCs reduce oxidative stress via suppression of ROS and RNS generation, and transferring healthy mitochondria to damaged cells [90]. Moreover, MSCs have anti-inflammatory properties, suppress pyroptosis and alleviate blood–brain barrier leakage [90,91]. MSCs also regulate imbalances in autophagy, thus conferring neuroprotection against cerebral ischemic injuries [92]. In general, MSCs seem to be a promising treatment option for stroke due to their pleiotropic effects.

## 6. Limitations of Current Antioxidant Therapy in Stroke

A trend has emerged, in which compounds with promising results in pre-clinical trials fail in clinical trials. There are several potential reasons for this. An obvious reason could be that an invalid compound or combination of compounds have been studied [93]. Another potential explanation could be that, although appropriate compounds were tested, they were tested in inadequate doses or durations [93]. The optimal dose for antioxidant therapy remains controversial, because the most effective doses are not known. For example, 800 units of vitamin E and 500 mg of vitamin C have been suggested to be the effective threshold dose; however, doses greater than 400 units a day have been suggested to increase mortality [94]. Another possible reason is the lack of measures of oxidation, without which it is not known whether sufficient amounts of antioxidants are absorbed or biologically active [95]. Finally, most trials were conducted with patients with established pathology, and since oxidation can occur through multiple pathways, different antioxidants with different targets can be combined to study the effects on clinical outcomes [95].

The lack of positive results in clinical trials does not necessarily disprove the important role of treating oxidative stress using antioxidants. Rather, these trials challenge us to better design antioxidant trials in the future and focus on using the best antioxidants at the right doses, in the right population and for the optimal duration.

## 7. Conclusions

Both mitochondrial dysfunction and massive inflammatory response are strongly associated with cerebrovascular diseases. Oxidative stress thus plays a major role in the pathophysiology of stroke and other brain-related pathologies. Highly reactive species have short half-lives; therefore, they react with the adjacent structures with no specific preference to any kind of macromolecules. Damage caused by oxidative stress is, thus, hard to predict. Targeting oxidative stress for the treatment of stroke is a challenge for the current research. Multimodal therapy, as well as innovative techniques such as therapy using nanoparticles or stem cells, are potential ways to minimize the harmful effect of oxidative stress.

This review was written using published articles that examined various antioxidant treatment strategies in stroke. An electronic database search of SpringerLink, ScienceDirect and PubMed/Medline was conducted with the search terms Stroke AND Antioxidants, (Stroke OR “Ischemic Stroke” OR “Hemorrhagic Stroke”) AND (Antioxidants OR “Antioxidant Therapy” OR “Antioxidant Treatment”), “Intracerebral Hemorrhage” AND (Antioxidants OR “Antioxidant Therapy” OR “Antioxidant Treatment”), and “Subarachnoid Hemorrhage” AND (Antioxidants OR “Antioxidant Therapy” OR “Antioxidant Treatment”). We excluded all non-English papers. Articles were excluded if they were unrelated to the review’s topic. Our goal was not to include all compounds used in stroke antioxidant treatment, but, rather, to provide a comprehensive review of the types of therapeutic approaches available, with examples of compounds used for each given type.

## Figures and Tables

**Figure 1 antioxidants-10-01886-f001:**
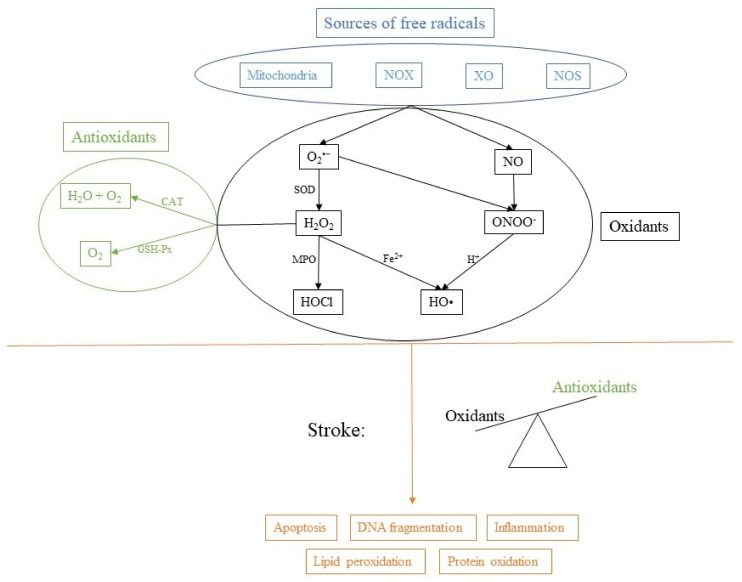
Sources of free radicals and consequences of imbalance. NADPH oxidase (NOX), xanthine oxidase (XO), nitric oxide synthase (NOS), superoxide dismutase (SOD), myeloperoxidase (MPO), catalase (CAT) and glutathione peroxidase (GSH-Px).

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
