# Peer review of "Oxidative Stress in the Brain: Basic Concepts and Treatment Strategies in Stroke"

_antioxidants, 2021, doi:10.3390/antiox10121886_

Round 1

Reviewer 1 Report

This is a useful and comprehensive review of oxidative stress as a source of damage in both hemorrhagic and ischemic stroke.

In addition to a nice discussion of the role of free radicals in stroke, the authors provide a useful Table 1 that highlights antioxidants discussed in the paper.

From  a clinical perspective, I think Table 1 could be improved significantly.

1) Indicate whether drug was studied pre-clinically or clinically

2) Indicate whether drug was "effective" in trial

3) Which drugs have been abandoned based on unfavorable results and which are still under consideration or evaluation?

In other diseases such as ALS that may have a basis in oxidative stress, a trend has emerged that drugs that are effective in pre-clinical trials fail in clinical trials.  Does such a trend exist in trials for stroke?

Reviewer 2 Report

This is an interesting overview on the basic concepts of oxidative stress and treatment strategies targeting oxidative stress in ischemic and hemorrhagic stroke.

Since this is a rather extensive topic, it would be helpful to specify the strategy used for the inclusion of studies in this review article (search methods and inclusion/exclusion criteria, criteria used to evaluate the quality of the evidence … see for example https://doi.org/10.1093/jnci/89.1.6). Outcomes used to evaluate the efficacy of stroke treatment should always be described and included in Table 1 as a separate column. Further, it would be of interest to the readers to include a translational perspective and shortly describe results of clinical studies and discuss potential reasons for the frequent failure in clinical trials.

Minor:

Compared to the ischemic stroke section, the hemorrhagic stroke section is very short, please give reasons for this (less evidence?).

The term cerebrovascular accidents is not common any more and should be replaced.

Round 2

Reviewer 1 Report

The authors have addressed my concerns and I have no further comments.

Reviewer 2 Report

The authors thoroughly revised the manuscript and addressed all points of criticism.